# Evolution of the Antigenic Landscape in Children and Young Adults with COVID-19 and MIS-C

**DOI:** 10.3390/vaccines12060638

**Published:** 2024-06-07

**Authors:** Lorenza Bellusci, Gabrielle Grubbs, Shaimaa Sait, Katherine W. Herbst, Juan C. Salazar, Surender Khurana

**Affiliations:** 1Division of Viral Products, Center for Biologics Evaluation and Research (CBER), FDA, Silver Spring, MD 20871, USA; 2Division of Pediatric Infectious Diseases, Connecticut Children’s, Hartford, CT 06106, USA; kherbst@connecticutchildrens.org (K.W.H.); jsalaza@connecticutchildrens.org (J.C.S.); 3Departments of Pediatrics and Immunology, School of Medicine, University of Connecticut, Farmington, CT 06030, USA

**Keywords:** SARS-CoV-2, children, COVID-19, MIS-C, vaccination

## Abstract

There is minimal knowledge regarding the durability of neutralization capacity and level of binding antibody generated against the highly transmissible circulating Omicron subvariants following SARS-CoV-2 infection in children with acute COVID-19 and those diagnosed with multisystem inflammatory syndrome in children (MIS-C) in the absence of vaccination. In this study, SARS-CoV-2 neutralization titers against the ancestral strain (WA1) and Omicron sublineages were evaluated in unvaccinated children admitted for COVID-19 (*n* = 32) and MIS-C (*n* = 32) at the time of hospitalization (baseline) and at six to eight weeks post-discharge (follow-up) between 1 April 2020, and 1 September 2022. In addition, antibody binding to the spike receptor binding domain (RBD) from WA1, BA.1, BA.2.75, and BA.4/BA.5 was determined using surface plasmon resonance (SPR). At baseline, the children with MIS-C demonstrated two-fold to three-fold higher binding and neutralizing antibodies against ancestral WA1 compared to those with COVID-19. Importantly, in children with COVID-19, the virus neutralization titers against the Omicron subvariants at six to eight weeks post-discharge reached the same level as those with MIS-C had at baseline but were higher than titers at 6–8 weeks post-discharge for MIS-C cases. Cross-neutralization capacity against recently emerged Omicron BQ.1, BQ.1.1, and XBB.1 variants was very low in children with either COVID-19 or MIS-C at all time points. These findings about post-infection immunity in children with either COVID-19 or MIS-C suggest the need for vaccinations in children with prior COVID-19 or MIS-C to provide effective protection from emerging and circulating SARS-CoV-2 variants.

## 1. Introduction

The coronavirus disease 2019 (COVID-19) pandemic caused by the widespread circulation of SARS-CoV-2 had a major impact on the global population. Even though the World Health Organization declared the end of the pandemic in May 2023 [1], there is still a high rate of virus transmission and hospitalization in the US [2]. Since its first appearance in late 2021, the Omicron variant, which contains a large number of mutations in the spike protein, has allowed it to escape prior immunity and therefore become the dominant circulating strain globally [3,4]. To date, Omicron is still evolving, generating more transmissible subvariants capable of escaping pre-existing immunity [5]. In 2023, the BQ.1.1 and XBB.1 variants became globally prevalent. They share a common substitution (R446T) that induces resistance to therapeutic antibodies [6]. Moreover, recent studies in adults showed that BQ.1.1 and XBB.1 were able to escape the humoral immunity response induced by prior SARS-CoV-2 infection or mRNA vaccination [5,7]. In contrast, limited information is available regarding the humoral immunity against currently circulating Omicron subvariants in children and young adults. Moreover, the rate of testing for young people remains low, leading to an underestimation of the true incidence of infection [8]. SARS-CoV-2 infection in children and young adults is usually mild or asymptomatic. However, children are susceptible to a severe form of COVID-19 and a severe inflammatory disorder called multisystem inflammatory syndrome in children (MIS-C) that presents at four to six weeks post-SARS-CoV-2 infection. As of November 2023, the Centers for Disease Control and Prevention (CDC) reported 9604 cases of children affected by MIS-C and 79 related deaths [9]. Common manifestations of the disease include abdominal pain, vomiting, diarrhea, skin rash, conjunctival injection, and hypotension [9,10]. Furthermore, in addition to these symptoms, children affected by MIS-C can present with higher rates of cardiac dysfunction, shock, myocarditis, coronary artery dilation, or aneurysm, and acute kidney injury [11].

Numerous studies have shown the importance of vaccination as a tool to protect people against the severe form of disease caused by the SARS-CoV-2 infection [12,13,14,15]. Unfortunately, the vaccination rate of children in the US remains very low (Figure 1 and https://covid.cdc.gov/covid-data-tracker/#vaccine-delivery-coverage, accessed on 16 March 2024), and the rates among children previously infected with SARS-CoV-2 are even lower, putting them at higher risk of infection from the currently circulating and emerging Omicron subvariants [2,16,17]. Moreover, limited knowledge is available regarding antibody-mediated immunity induced in children with severe acute COVID-19 versus MIS-C, both at the time of presentation and after disease resolution, against the currently circulating SARS-CoV-2 Omicron variants. Additionally, minimal information is available on the evolution of the antigenic landscape between COVID-19 and MIS-C over time. Hence, it is important to assess the humoral responses in children and young adults with prior COVID-19 or MIS-C against the currently circulating Omicron subvariants. This could provide crucial information to identify and implement better approaches to help fight the disease in children and young adults. Therefore, the aim of our study was to evaluate the capacity of neutralizing antibodies induced following SARS-CoV-2 infection, either during acute COVID-19 or those induced by MIS-C against the SARS-CoV-2 ancestral WA1 strain, the earlier circulating Alpha and Delta strains, Omicron BA.1, BA.2, BA.2.75, BA.4/BA.5, and newly circulating BQ.1, BQ.1.1, and XBB.1 subvariants, at the time of hospitalization and their evolution at six to eight weeks post-discharge in children and young adults. 

## 2. Methods

### 2.1. Ethics Statement

The collection of samples from children received ethics approval from the Connecticut Children’s Institutional Review Board (IRB) #21-004. Informed consent and assent, when appropriate, were obtained from all participants and/or their parents/legal guardians. All methods and procedures were approved by the IRB and carried out in accordance with the IRB’s guidelines and regulations. 

The study at CBER, FDA, was conducted with de-identified samples. Antibody assays were performed with approval from the US Food and Drug Administration’s Research Involving Human Subjects Committee (FDA-RIHSC) under exemption protocol ‘252-Determination-CBER-2020-08-19’. This study complied with all relevant ethical regulations for working with human participants. All assays performed fell within the permissible usages of the original informed consent.

### 2.2. Study Design

Biospecimens were obtained from subjects enrolled at a pediatric hospital between May 2020 and August 2022. Subjects were eligible for enrollment if they were <21 years of age and hospitalized with a diagnosis of either COVID-19 per SARS-CoV-2 antigen, PCR, or at-home test or MIS-C per the Center for Disease Control’s criteria. To ensure correct diagnostic group assignment, cases were reviewed by three pediatric specialists, and agreement on the primary diagnosis was reached. Baseline blood was collected at enrollment and again at six to eight weeks post-enrollment. Demographic, health history, current symptoms, vaccine history, in-patient treatments, and diagnostic testing results were collected at baseline, and vaccine history and current health were collected at follow-up. Sera was isolated from participant blood samples by resting the blood for approximately 30 min after collection to allow for coagulation. The non-coagulated part of the blood was transferred into a 15 mL tube and centrifuged at 1000× *g* for 15 min. The supernatant was then transferred into a new tube and centrifuged at 1000× *g* for 5 min. The supernatant was aliquoted into 200 µL cryovials and stored at −80 °C until transferred to participating laboratories. Plasma was isolated from participant blood samples by diluting blood 1:1 with Hanks’ Balanced Salt Solution, then streaming into a purple Sepmate tube containing 12 mL of Ficoll-paque below the filter separation. The tube was then centrifuged at 1000× *g* for 10 min with no break, plasma aliquoted into 1 mL cryovials, and stored at −80 °C until transferred to participating laboratories.

The objective of this study was to investigate the antibody binding and neutralizing activity of serum/plasma in children < 21 years old hospitalized for COVID-19 or for MIS-C against circulating SARS-CoV-2 variants. We analyzed samples from a total of 64 subjects included in this study, i.e., *n* = 32 who have been hospitalized for COVID-19 and *n* = 32 who received a diagnosis of MIS-C. 

### 2.3. Lentivirus Pseudovirion Neutralization Assay (PsVNA)

Samples were evaluated in a qualified SARS-CoV-2 pseudovirion neutralization assay (PsVNA) using SARS-CoV-2 WA1/2020, Alpha, and Delta strains, Omicron subvariants BA.1, BA.2, BA.2.75, BA.4/BA.5, and the circulating BQ.1, BQ.1.1, and XBB.1. The mutations in spike protein of these Omicron subvariants are shown in Appendix A. SARS-CoV-2 neutralizing activity measured by PsVNA correlates with PRNT (plaque reduction neutralization test with authentic SARS-CoV-2 virus) in previous studies [18,19]. 

Neutralization assays were performed as previously described [18,19,20,21]. Briefly, 50 µL of SARS-CoV-2 spike pseudovirions (counting ~200,000 relative light units) were pre-incubated with an equal volume of medium containing serial dilutions (starting at 1:10) of all samples at room temperature for 1 h. Then, 50 µL of virus–antibody mixtures were added to 293T-ACE2-TMPRSS2 cells [22] [10^4^ cells/50 μL] in a 96-well plate. The input virus with all SARS-CoV-2 strains was the same (2 × 10^5^ relative light units/50 µL/well). After 3 h of incubation, fresh medium was added to the wells. Cells were lysed 24 h later, and luciferase activity was measured using the One-Glo luciferase assay system (Promega, Madison, WI, USA). The assay of each sample was performed in duplicate, and the 50% neutralization titer was calculated using Prism 9 (GraphPad Prism Software). The limit of detection for the neutralization assay is 1:20. Two independent biological replicate experiments were performed for each sample, and the variation in PsVNA50 titers was <10% between replicates.

### 2.4. Antibody Binding Kinetics to SARS-CoV-2 RBD by Surface Plasmon Resonance (SPR)

Purified recombinant SARS-CoV-2 spike receptor binding domain (RBD) of WA1 and Omicron strains expressed in HEK293 cells were purchased from Sino Biologicals (Beijing, China). Steady-state equilibrium binding in serum/plasma samples was monitored at 25 °C using a ProteOn SPR (BioRad, Hercules, CA, USA). The SARS-CoV-2 RBD proteins were captured on a Ni-NTA sensor chip with 500 resonance units (RU) in the test flow channels [19]. 

Serial dilutions (10-, 30-, and 90-fold) of freshly prepared sample diluted in BSA-PBST buffer (PBS pH 7.4 buffer with Tween-20 and bovine serum albumin) were injected at a flow rate of 50 µL/min (120 sec contact duration) for association, and disassociation was performed over a 600-second interval. Responses from the protein surface were corrected for the response from a mock surface and for responses from a buffer-only injection. SPR was performed with serially diluted samples in this study. Total antibody binding was calculated with BioRad ProteOn manager software (version 3.1). All SPR experiments were performed twice, and the researchers performing the assay were blinded to sample identity. The variations for duplicate runs of SPR were <5%. The maximum resonance units (Max RU) shown in the figures were the calculated RU signal for the 10-fold dilution sample. 

### 2.5. Statistical Analysis

Descriptive statistics were performed to determine the geometric mean titer values and were calculated using GraphPad Prism (9.3.1 version). Cohort characteristics were compared using the Mann–Whitney U test, Fisher’s Exact test, or Fisher–Freeman–Halton Exact test, with the *p*-value adjusted via the Bonferroni method and calculated in SPSS 28.0.1.1 (IBM Corporation, Armonk, NY, USA). All experimental data to compare differences between COVID-19 and MIS-C groups were analyzed by ANOVA using the Kruskal–Wallis test for multiple comparisons calculated in GraphPad Prism (9.3.1 version). Within each disease category, for paired baseline and follow-up samples for COVID-19 or MIS-C, the comparison was performed using the Friedman test as shown in Appendix A. To ensure the robustness of the results, absolute measurements were log2-transformed before performing the analysis. 

Samples were allocated randomly to each test group and tested blindly (the researcher was blinded to sample identity) to minimize selection bias or detection bias. All samples and data were used for the analysis and are presented in this study.

### 2.6. Antigenic Landscape and Cartography

Antigenic cartography analysis was performed using the Racmacs package in R software (version 4.1.2). Multi-dimensional antigenic landscape maps were developed using the Rossler method, which measures the antigenic distance as well as the magnitude of antibody binding kinetics and was constructed as described before [23,24,25]. 

To evaluate the robustness of our findings and account for potential variation in antigenic cartography, bootstrap analysis was conducted with 1000 bootstrap repeats and 100 optimizations per repeat, as described before [25], using the R package (Racmacs). The bootstrap analysis incorporated noise by adding normally distributed values to both the titers and antigen reactivity. The standard deviation of the noise added to the neutralization titers was 0.7, while the standard deviation for the noise added to antigen reactivity was 0.7.

### 2.7. Code Availability

Antibody titers were calculated using Prism 9.3.1 (GraphPad Software). Statistical analyses were performed using Prism 9.3.1 (GraphPad Software). No new code was generated.

## 3. Results

### 3.1. Study Cohort

Limited knowledge is available on the evolution of the antigenic landscape in unvaccinated SARS-CoV-2-infected children hospitalized for COVID-19 vs. MIS-C against circulating SARS-CoV-2 variants. Therefore, we analyzed samples from a total of 64 children (<20 years old) who were hospitalized between May 2020 and August 2022 for SARS-CoV-2 neutralizing antibodies and RBD binding antibodies using surface plasmon resonance (SPR) against WA1 and Omicron subvariants (Figure 2). Pediatric samples were divided into the following two categories: *n* = 32 children who have been hospitalized with a diagnosis of COVID-19 per SARS-CoV-2 antigen, PCR, or at-home test (COVID-19), and *n* = 32 children hospitalized with a diagnosis of MIS-C per the Center for Disease Control’s criteria (MIS-C). For each category, baseline blood was collected at enrollment and upon hospitalization. Of those, 19 subjects (11 COVID-19, 8 MIS-C) had follow-up samples available at six to eight weeks post-enrollment. The majority of subjects were non-Hispanic males hospitalized at a median age of 10 years (interquartile range 6–15 years) (Table 1).

### 3.2. Neutralizing Antibodies in Children with COVID-19 vs. MIS-C against Circulating SARS-CoV-2 Variants 

Antibody neutralization activity of children and adolescents’ samples was evaluated against the infecting ancestral SARS-CoV-2 WA1, Alpha, Delta, and Omicron BA.1 strains that circulated during May 2020 and August 2022. We also tested the capacity of these antibodies to neutralize different Omicron subvariants that emerged subsequently, including BA.2, BA.2.75, BA.4/BA.5, and highly transmissible BQ.1, BQ.1.1, and XBB.1 subvariants, using the pseudovirion neutralization assay (PsVNA). SARS-CoV-2 neutralizing activity measured by PsVNA correlates with PRNT (plaque reduction neutralization test with authentic SARS-CoV-2 virus), as described in previous studies [18,19]. The mutations in the spike protein of these Omicron subvariants compared with ancestral WA1 are shown in Appendix A.

At baseline, the COVID-19 cohort showed high variability in titers of neutralizing antibodies (PsVNA50 titers; dilution of sample that achieved 50% virus neutralization) against the ancestral WA1, with values ranging between 10 and 2138 (geometric mean titer (GMT) of 55) and seropositivity rate (samples with PsVNA50 titers ≥ 60) of 47% (Figure 3A). Neutralizing titers against both the Alpha and Delta variants showed a similar trend of variability, with PsVNA50 titers ranging between 10 and 2255 and between 10 and 2748, respectively. Conversely, a very weak response was observed against all the Omicron subvariants, with the lowest neutralizing titers recorded against the circulating highly transmissible BQ.1, BQ.1.1, and XBB.1 subvariants with a seropositivity rate of 0–3% and GMT of 11–12 only. At 6–8 weeks of follow-up, the COVID-19 cohort showed an increase in neutralizing antibodies. For the 11 COVID-19 patients with paired baseline and follow-up samples, PsVNA50 titers against the WA1 strain ranged between 26 and 907 with a GMT of 138 and showed an increase of 1.2-fold in neutralizing antibodies compared to the corresponding baseline, and seropositivity increase from 78% to 89% (Appendix A). A similar rise in titers of 1.4- and 1.1-fold was observed against the Alpha and Delta strains, respectively. Unlike the immune response against the earlier circulating strains, only a slight increase in neutralization titers against some Omicron variants was observed at follow-up compared with baseline, with no or minimal seropositivity (Figure 3A). 

Among the MIS-C cohort, baseline PsVNA50 titers against the WA1, Alpha, and Delta strains were higher compared with the COVID-19 cohort’s levels, with GMT values of 169, 161, and 93, respectively, and neutralizing antibodies were shown to be protective (PsVNA50 titers ≥ 60) against these variants, with a seropositivity ranging between 69% and 81% (Figure 3B). At the time of hospitalization, PsVNA50 titers against WA1, as well as Alpha and Delta strains, were 2.5 times higher for the MIS-C cohort compared with COVID-19 cases at baseline (Figure 3C). The immune response against the Omicron subvariants was very low, similar to the COVID-19 baseline cohort. However, a decline in neutralizing antibodies was observed at 6–8 weeks follow-up, opposite to what was observed for the COVID-19 cohort. For the nine paired MIS-C patients, the neutralization titers decreased over the weeks by 1.6 folds against the WA1 strains and by 1.9 and 2.5 folds against the Alpha and Delta variants, respectively, reducing the seropositivity for Alpha to 43% and for Delta to 29% (Appendix A). PsVNA50 titers against the Omicron subvariants, already low at the time of hospitalization, remained low at 6 to 8 weeks of follow-up (Figure 3B and Appendix A). 

The trend observed for neutralizing antibody titers among both the COVID-19 and MIS-C cohorts was confirmed by the antibody binding against the RBD of WA1, BA.1, BA.2.75, and BA.4/BA.5 spike proteins using SPR (Figure 4). The RBD-binding antibodies from samples of children hospitalized for acute COVID-19 increased over 6 to 8 weeks, similar to the observation of neutralization titers (Figure 4A). For 11 paired COVID-19 patients, the binding antibodies against RBD derived from WA1 or Omicron variants increased by 1.1- and 1.2-fold (Appendix A). For the MIS-C cohort, the trend was similar to the pseudovirus neutralization results, with a decline in RBD-binding antibodies at 6–8 weeks follow-up compared with baseline antibody titers (Figure 4B). For the nine paired MIS-C patients, antibodies binding to BA.2.75 and to BA.4/BA.5 showed the highest decrease (6.4-fold) over the 6–8 weeks (Appendix A). We observed a similar pattern of drop in RBD binding antibody reactivity in SPR, similar to what was observed with the PsVNA50 titers for these MIS-C samples. A comparison between the baselines of the two different cohorts shows the antibody binding was higher for the MIS-C group compared with COVID-19 cohort at the time of hospitalization against all the RBDs tested (Figure 4C).

### 3.3. Neutralizing Antibody Landscapes in Children with COVID-19 vs. MIS-C

We performed two-dimensional antigenic cartography to determine the antigenic relationship of the neutralizing antibodies against the ancestral WA1 and different SARS-CoV-2 Omicron variants across the two cohorts at baseline and follow-up time points (Figure 5). In the acute COVID-19 group, both Omicron and Delta variants were very close to the WA1 and Alpha strains at baseline. Over the 6–8-week follow-up, the anti-Omicron variant titers showed a larger distance from the titers against the early strains, with XBB.1 being the most distant (Figure 5A). Among the MIS-C baseline, the Omicron subvariants, as well as Delta, showed more distance from WA1 compared to the COVID-19 baseline. However, at 6–8 weeks of follow-up, the different strains became closer due to a decrease in neutralizing titers in the MIS-C cohort (Figure 5B). Thus, early during infection, the COVID-19 baseline and MIS-C 6–8-week groups shared a similar and closer antigenic relationship between Omicron subvariants and WA1. Conversely, for both COVID-19 6–8 weeks and MIS-C baseline groups, the Omicron subvariants had substantially diversified away from WA1.

We evaluated the antigenic landscape for the COVID-19 and MIS-C children’s cohorts to quantify the magnitude of this antigenic relationship (Figure 6). We analyzed the three-dimensional antigenic landscape of the baseline and 6–8-week groups of each child cohort based on the neutralization titers presented in Figure 3. As expected, the slope of the landscape against WA1, Alpha, and Delta of samples from both COVID-19 6–8 weeks and MIS-C baseline groups was very similar, as was their magnitude (Figure 6A,B). Moreover, similar slopes of the antigenic landscape against the Omicron subvariants were shared by the two groups as well. At baseline, the MIS-C group showed a higher magnitude than the COVID-19 group against earlier circulating strains (WA1, Alpha, and Delta) with a steep slope against Omicron variants (Figure 6C). In addition, as we already observed, over the weeks, neutralizing antibodies from MIS-C 6–8-week group samples decreased; this trend was confirmed by the slopes of the antigenic landscape against the earlier circulating variants, which were very much alike to the slope of the antigenic landscape of the COVID-19 baseline group (Figure 6C,D).

## 4. Discussion

SARS-CoV-2 still represents a global threat that challenges the updating of vaccines and therapies, with the evolution of new Omicron subvariants that show increased transmissibility and antibody escape mutations. Since in children, SARS-CoV-2 infections and related hospitalizations are less common compared with adults, they constitute an underrepresented and understudied population. However, they are susceptible to severe manifestations of COVID-19. Moreover, a severe inflammatory post-infectious syndrome, referred to as MIS-C, has been associated with SARS-CoV-2 infection in children. Furthermore, vaccination rates in children, especially those with prior SARS-CoV-2 infection, remain extremely low [13,14,15]. Therefore, it is important to study how the immune response induced by either COVID-19 or the consequent MIS-C evolves over time in children and if the antibodies produced will be cross-neutralizing against the currently circulating highly contagious SARS-CoV-2 BQ.1, BQ.1.1, and XBB.1 subvariants. This, could help better understand both the best therapeutic and the vaccination approach for this age population. 

In this pediatric study, children who have been hospitalized following acute COVID-19 demonstrate a high variability in neutralizing antibodies against the ancestral WA1 infecting strain as well as against Alpha and Delta variants in the early stages of the disease that increased in titers after 6 to 8 weeks. In contrast, we did not observe any relevant cross-reactivity against the Omicron subvariants in the baseline; moreover, little or no increase in titers was observed after 6 to 8 weeks. Children hospitalized with COVID-19 did not have enough time to produce a strong antibody response. Furthermore, the acute SARS-CoV-2 infection could have contributed to the suppression of the antibody response. After 6 to 8 weeks of COVID-19 resolution, together with the improvement of the health condition, the children had time to develop neutralizing antibodies. 

In contrast, children hospitalized for MIS-C showed a higher antibody response in the early stages of the disease. In addition, the PsVNA50 titers against WA1, as well as Alpha and Delta variants, were 2.5 times higher for the MIS-C cohort at baseline compared with the neutralizing titers from samples of children collected at baseline from COVID-19 hospitalized children. In contrast to the COVID-19 cohort, MIS-C antibodies decreased over time. These latest results could be explained since MIS-C is a post-infectious complication that usually affects children after 4 to 6 weeks from a previous SARS-CoV-2 infection. Thus, children had sufficient time to produce an anti-SARS-CoV-2 antibody response. Unfortunately, we observed that neutralizing antibodies against the ancestral WA1 or Alpha strain generated after both COVID-19 and MIS-C are not sufficient to cross-neutralize the currently circulating highly contagious Omicron BQ.1, BQ.1.1, and XBB.1. Limitations of this study include sera and plasma derived from a limited number of children. The findings in this study should be confirmed in future longitudinal studies with a larger cohort of children. Moreover, specimens were obtained from children and young adults infected with SARS-CoV-2 between May 2020 and August 2022, during which the circulating SARS-CoV-2 strains were WA1, Alpha, Delta, and Omicron BA.1. Therefore, it will be important to collect samples from children and young adults recently infected with currently circulating Omicron subvariants in order to assess the immune response and durability of neutralizing antibodies against emerging SARS-CoV-2 variants. This will help assess appropriate public health approaches to curtail SARS-CoV-2 transmission and disease in children.

## 5. Conclusions

Control of SARS-CoV-2 infection, transmission, or disease in children is crucial to preventing the long-term consequences of viral infection in this understudied and underrepresented population. This study demonstrates that SARS-CoV-2 infection in children with either COVID-19 or MIS-C at the time of hospitalization or at the 6–8-week follow-up convalescent phase contains minimal or no neutralizing antibodies against the currently circulating BQ.1, BQ.1.1, and XBB.1 subvariants that could influence the clinical outcome and transmission in children. Previous studies showed that vaccination in children induced the highest cross-neutralizing antibodies against the newly emerging Omicron subvariants, compared with children hospitalized either for COVID-19 or MIS-C [26]. Therefore, vaccination of both naïve and prior SARS-CoV-2-exposed children with updated SARS-CoV-2 vaccines can generate broadly cross-neutralizing antibodies against circulating and emerging SARS-CoV-2 variants, eventually providing more effective protection for children against disease caused by the currently circulating and emerging SARS-CoV-2 variants. 

## Figures and Tables

**Figure 1 vaccines-12-00638-f001:**
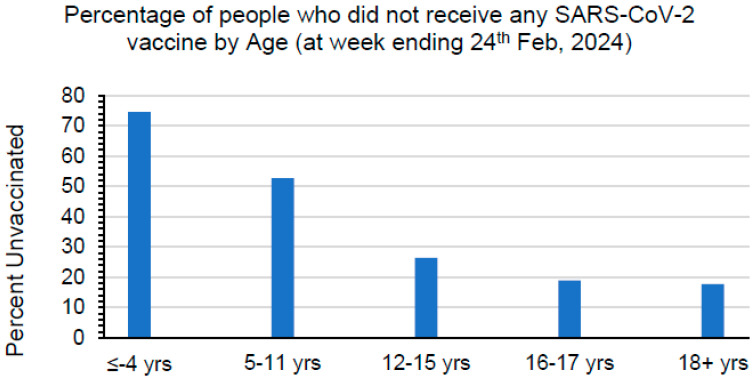
Percentage of people who remain unvaccinated with the SARS-CoV-2 vaccine by different age groups in the US (from 14 December 2020 to 24 February 2024), as per data from the US CDC (https://covid.cdc.gov/covid-data-tracker/#vaccine-delivery-coverage, accessed on 16 March 2024).

**Figure 2 vaccines-12-00638-f002:**
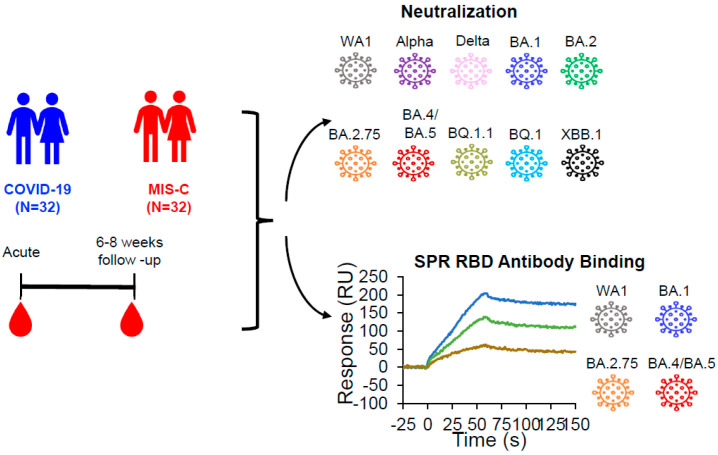
Study design of children following acute COVID-19, or MISC. Overview of the child cohort with acute COVID-19, or MIS-C. Each child’s sample was evaluated for neutralizing antibodies against ten SARS-CoV-2 strains in a pseudovirus neutralization assay and for binding antibodies against prototype WA1 and three Omicron RBD’s using surface plasmon resonance.

**Figure 3 vaccines-12-00638-f003:**
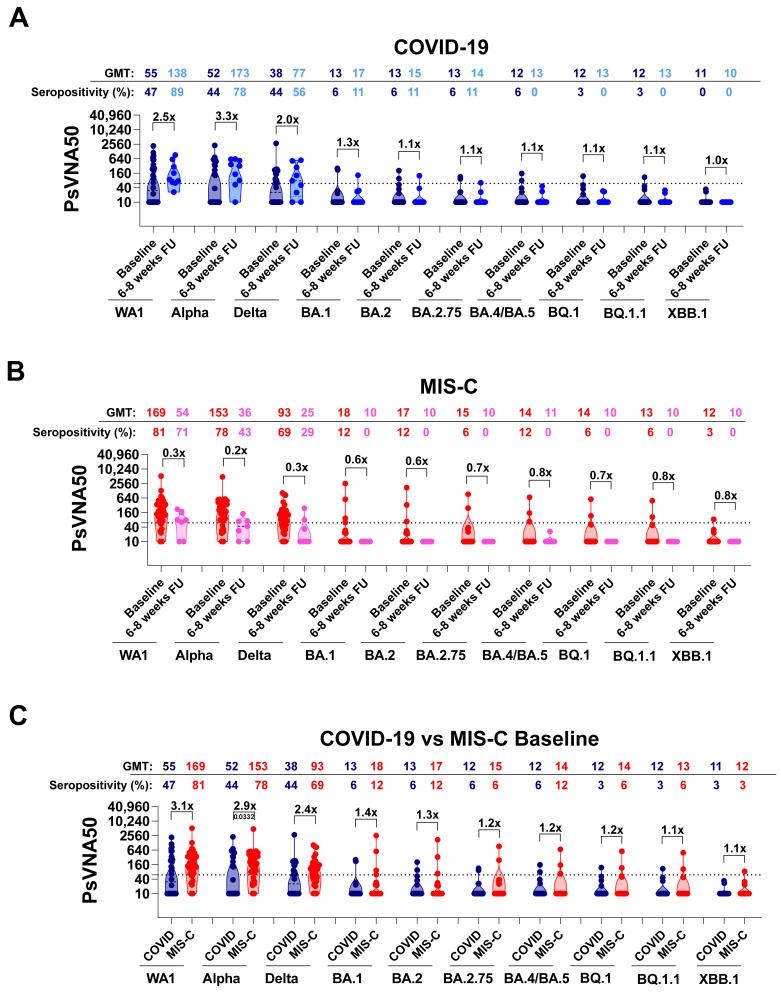
Neutralizing antibodies in children with COVID-19 vs. MIS-C against circulating SARS-CoV-2 variants. SARS-CoV-2 neutralization assays were performed by using pseudoviruses expressing the spike protein of WA1, Alpha, Delta, or the Omicron subvariants BA.1, BA.2, BA.2.75, BA.4/BA.5, BQ.1, BQ.1.1, and XBB.1 subvariants in 293-ACE2-TMPRSS2 cells. (**A**–**C**) SARS-CoV-2 neutralization titers were determined in each of the 64 children hospitalized with COVID-19 (*n* = 32) or MIS-C (*n* = 32) at the time of enrollment and after 6 to 8 weeks (COVID-19; *n* = 11 and MIS-C; *n* = 9). The assay of each sample was performed in duplicate to determine the 50% neutralization titer (PsVNA50). PsVNA50 titers for COVID-19 children (**A**) are shown in dark blue (baseline) and light blue (after 6–8 weeks), whereas PsVNA50 titers for MIS-C children (**B**) are shown in red (baseline) and pink (6–8 weeks). (**C**) PsVNA50 titers at the baseline of the COVID-19 cohort (*n* = 32) versus the baseline of the MIS-C cohort (*n* = 32) are represented. The heights of the bars and the numbers over the bars indicate the geometric mean titers (GMT), and the whiskers indicate 95% confidence intervals. The horizontal dashed line indicates the seropositive cut-off for the neutralization titers (PsVNA50 of 60). Percent seropositivity (%S) for each group was calculated as the number of seropositive samples divided by the total number of samples × 100 in the group. A fold change of 6–8 weeks follow-up compared with baseline samples is shown for PsVNA50 titers of each strain. The limit of detection for the neutralization assay is 1:20. Differences between the baseline vs. follow-up within each SARS-CoV-2 strain (for (**A**,**B**)) or between COVID-19 vs. MIS-C (**C**) were analyzed by ANOVA with Kruskal–Wallis test for multiple comparisons. Statistically significant *p*-values are shown.

**Figure 4 vaccines-12-00638-f004:**
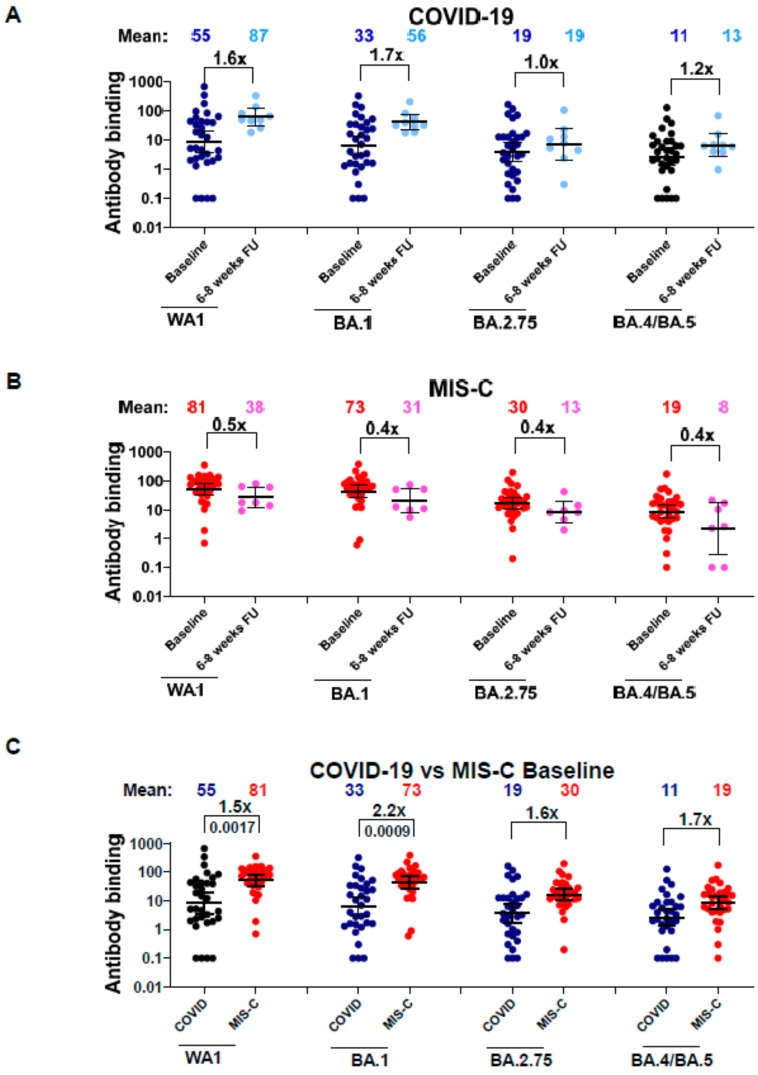
Binding antibodies in the serum/plasma of children with COVID-19 or MIS-C to the receptor binding domain of the ancestral SARS-CoV-2 WA1 strain and Omicron BA.1, BA.2.75, and BA.4/BA.5 variants. (**A**–**C**) Total antibody binding (determined by maximum resonance units, Max RU) of 1:10 diluted serum or plasma to purified WA1 spike RBD, BA.1 spike RBD, BA.2.75 spike RBD, and BA.4/BA.5 spike RBD was measured by SPR. Data are shown by disease groups for samples from children hospitalized for COVID-19 (*n* = 32) and for children admitted to the hospital for MIS-C (*n* = 32). (**A**) For the COVID-19 cohort, dark blue represents the baseline samples, whereas light blue represents the 6–8-week samples. (**B**) For the MIS-C cohort, red represents the baseline samples and pink represents the 6–8-week samples. The heights of the bars and the numbers over the bars indicate the mean antibody binding values, and the whiskers indicate 95% confidence intervals. A fold change of 6–8 weeks follow-up compared with baseline samples is shown for the RBD-binding antibodies of each strain. All SPR experiments were performed in duplicate, and the researchers performing the assay were blinded to sample identity. The variations for duplicate runs of SPR were <5%. The data shown are the average values of two experimental runs. The statistical significances for the baseline vs. follow-up within each SARS-CoV-2 strain (for (**A**,**B**)) or between COVID-19 vs. MIS-C (**C**) at baseline were analyzed by ANOVA with Kruskal–Wallis test for multiple comparisons. Statistically significant *p*-values are shown.

**Figure 5 vaccines-12-00638-f005:**
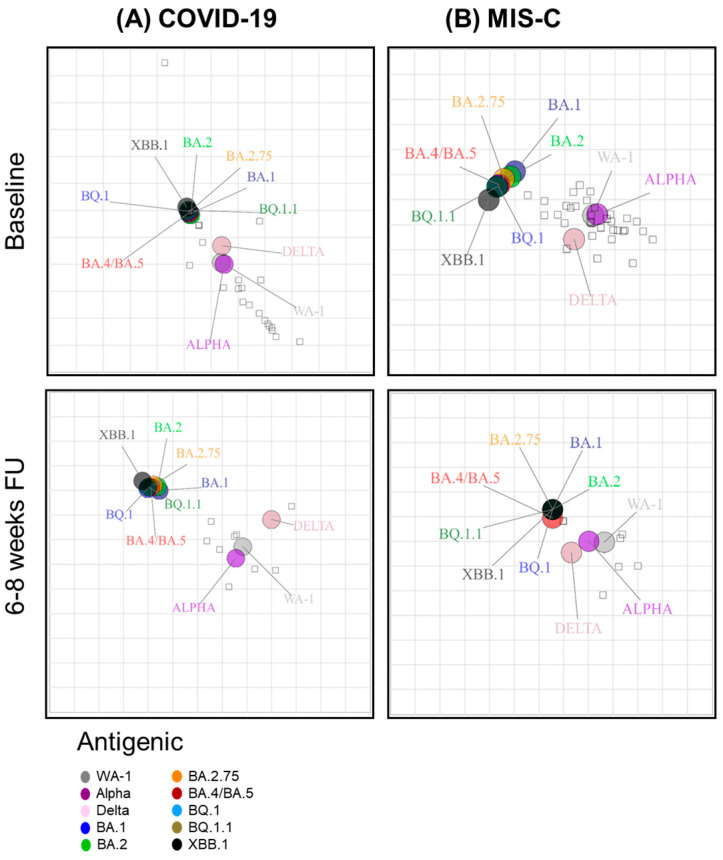
Antigenic cartography in children with COVID-19 vs. MIS-C. Individual antigenic maps were generated for each COVID-19 or MIS-C cohort ((**A**,**B**), respectively) at baseline and 6–8 weeks follow-up time points against SARS-CoV-2 WA1, Alpha, Delta, and the Omicron subvariants. Black diamonds correspond to each individual sera/plasma. One antigenic distance represented by each grid square corresponds to a twofold dilution of the neutralization assay.

**Figure 6 vaccines-12-00638-f006:**
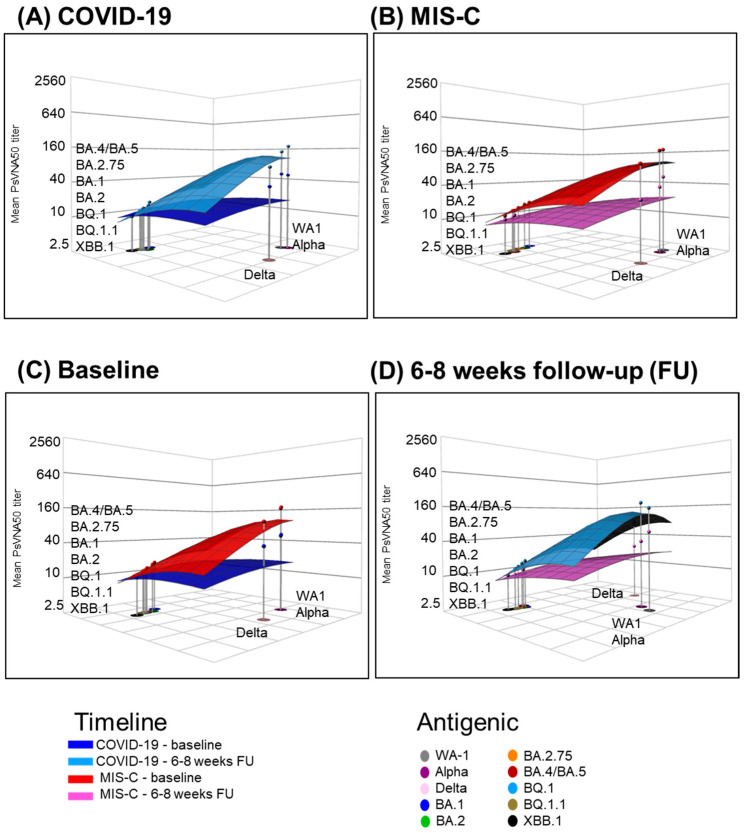
Neutralizing antibody landscapes in children with COVID-19 vs. MIS-C. The antigenic landscape was generated using SARS-CoV-2 neutralization titers against WA1, Alpha, Delta, and Omicron BA.1, BA.2, BA.2.75, BA.4/BA.5, BQ.1, BQ.1.1, and XBB.1 for the 64 children with either COVID-19 or MIS-C. (**A**) Neutralizing antibody landscapes for the samples from children hospitalized for acute COVID-19 (*n* = 32) at the time of enrollment (dark blue) and after 6 to 8 weeks (light blue). (**B**) Neutralizing antibody landscapes for the samples from children hospitalized for MIS-C (*n* = 32) at the time of enrollment (red) and after 6 to 8 weeks (pink). (**C**) Comparison between the neutralization antibody landscapes for the baseline samples of the two cohorts. (**D**) Comparison between the neutralizing antibody landscapes of the 6–8-week samples from the two different cohorts. The base x- and y-axes of each landscape map represent antigenic cartography between WA1 and variants, with colored points representing the locations of each SARS-CoV-2 strain. The grid squares (1 antigenic unit) correspond to a twofold change in the neutralization assay. The vertical *z*-axis corresponds to the neutralization titer on the log2 scale. The overall neutralizing antibody landscape for each pediatric group was constructed by fitting individual neutralizing antibody landscapes for each child’s serum sample for that group against the respective SARS-CoV-2 strain. The landscapes are color-coded according to patient categories (COVID-19 or MIS-C). The magnitude of neutralization titers (PsVNA50 GMTs) for children either following COVID-19 infection or MIS-C against each SARS-CoV-2 strain is shown by impulses connected for each SARS-CoV-2 variant on the *z*-axis in the landscape, and the filled-circle symbols are color-coded by each patient category group. A Mann–Whitney test in the R package was used to determine the significance of the difference between patient cohorts for each age group, and the significant *p*-values are shown. The tests were two-sided tests.

**Table 1 vaccines-12-00638-t001:** Study cohort characteristics.

Characteristic	COVID-19 (*n* = 32)	MIS-C (*n* = 32)	*p*-Value
Age (months)	113 (44–198)	116 (69–154)	0.50
Male	17 (53%)	19 (59%)	0.80
Race			0.15
Black	5 (16%)	12 (38%)	
White	13 (41%)	5 (16%)	
Other Race	11 (34%)	12 (37%)	
>1 Race	1 (3%)	3 (9%)	
Prefer not to answer	2 (6%)	-	
Hispanic	11 (34%)	12 (37%)	1.00

Data reported as median (IQR) or n (%) between group comparisons made using Mann–Whitney U, Fisher’s Exact test, or Fisher–Freeman–Halton Exact test (*p*-value adjusted via the Bonferroni method).

## Data Availability

All data are shown in the manuscript figures and Appendix A.

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
