# Peer review of "Evolution of the Antigenic Landscape in Children and Young Adults with COVID-19 and MIS-C"

_vaccines, 2024, doi:10.3390/vaccines12060638_

Round 1

Reviewer 1 Report

Comments and Suggestions for Authors

In this study, SARS-CoV-2 neutralization titers against the ancestral strain (WA1) and Omicron sub-lineages were evaluated in unvaccinated children admitted for COVID-19 and MIS-C at the time of hospitalization (baseline) and at six to eight weeks post-discharge (follow-up) between April 1, 2020, and October 1, 2022. In addition, antibody binding against spike proteins from WA1, BA.1, BA.2.75 and BA.4/BA.5 was determined using surface plasmon resonance (SPR). At baseline, the children with MIS-C demonstrated two- to three-fold higher binding and neutralizing antibodies against ancestral WA1 compared to those with COVID-19. Importantly, in children with COVID-19, the virus neutralization titers against the Omicron sub-variants at six to eight weeks post-discharge reached the same level as those with MIS-C at baseline but were higher than titers at 6–8 weeks post-discharge for MIS-C cases. Cross-neutralization capacity against recently emerged Omicron BQ.1, BQ.1.1 and XBB.1 variants was very low in children with either COVID-19 or MIS-C at all time points. These findings about post-infection immunity in children with either COVID-19 or MIS-C suggest the need for vaccinations in children with prior COVID-19 or MIS-C to provide effective protection from emerging and circulating SARS-CoV-2 variants.

The method and data used in this study are well documented by authors and I believe that this article will be useful to the readers of this international journal. I find the article to be interesting and below are minor revisions recommended to the authors.

Major comment

The introduction part of this work must be well expanded. I am not satisfied with how the introduction was structured. The authors must clearly state the motivation of this work, the existing literature in this research direction and the general contributions of this research.

Figure 3 should be better presented as line plot to show the confidence interval appropriately.

In section 4.5, the statistical analysis description presented by the authors is very confusing as authors are comparing parametric method with non-parametric method. What is the justification for this?

This work lacks conclusion. The authors must clearly state limitations, perhaps future directions, and how the work can be improved. 

Comments on the Quality of English Language

Moderate editing of English language required

Author Response

We appreciate reviewer’s comments to improve our manuscript. We have addressed all of them in the revised manuscript. We have provided point-by-point responses to the reviewer comments.

Reviewer 1:

Major comment

The introduction part of this work must be well expanded. I am not satisfied with how the introduction was structured. The authors must clearly state the motivation of this work, the existing literature in this research direction and the general contributions of this research.

Response: The introduction has been expanded and restructured as per reviewer suggestion.

Figure 3 should be better presented as line plot to show the confidence interval appropriately.

Response: Old figure 3 has been revised to show GMT with 95 CI in the modified new figure 4. The data for matched paired samples from each child is shown in supplementary figure 2.

In section 4.5, the statistical analysis description presented by the authors is very confusing as authors are comparing parametric method with non-parametric method. What is the justification for this?

Response: We have clarified it the revised manuscript. All experimental data to compare differences between COVID-19 vs MIS-C groups were analyzed by ANOVA using Kruskal-Wallis test for multiple comparisons calculated in GraphPad Prism (9.3.1 version).

Within each disease category, for paired baseline and follow up samples of COVID-19 or MIS-C, the comparison was performed using the Friedman test as shown in supplementary figure S2.

This work lacks conclusion. The authors must clearly state limitations, perhaps future directions, and how the work can be improved.

Response: Limitations with future directions have been added to the discussion section. Conclusion section has been added at the end of discussion section.

Comments on the Quality of English Language

Moderate editing of English language required.

Response: Manuscript has been edited and proofread for English language.

Reviewer 2 Report

Comments and Suggestions for Authors

Minor revisions

Title and wherever else needed : Rephrase, since the age range (0-21years) includes not only children but young adults as well.

Figure 1A: Graph A of Figure 1 does not derive from the present results. Therefore, it could be either removed or moved to the Introduction if allowed by the journal instructions.

Figure 1A: The “SPR RBD Antibody Binding” graph of Part B of Figure 1 seems to present data about binding antibodies against BA.1, BA.2, and most probably BA.2.75. What are the variant selection criteria? What about data against the prototype WA1?

Table 1: The upper age limit (241 months) must be 20 instead of 21 years.

L. 180: What are the selection criteria for the targeted variants?

L. 342: Have pediatric specialists been involved in all the cases? To be clarified. (According to the CDC, in medical terms the age limit between children and adults is 17 years.)

Major revision

Overall: Each infecting variant is absolutely related to the produced specific antibodies and to the resulting cross reactions as well. What are the variants by which the research subjects were infected? In case there are no such details, it should be mentioned.

Author Response

We appreciate the reviewer’s comments to improve our manuscript. We have addressed all of them in the revised manuscript. We have provided point-by-point responses to the reviewer comments.

Reviewer 2:

Minor revisions

Title and wherever else needed: Rephrase, since the age range (0-21years) includes not only children but young adults as well.

Response: Title is revised as per suggestion to: Evolution of antigenic landscape in children and young adults with COVID-19 and MIS-C.

Figure 1A: Graph A of Figure 1 does not derive from the present results. Therefore, it could be either removed or moved to the Introduction if allowed by the journal instructions.

Response: Figure 1a has been moved to introduction as standalone Figure 1.

Figure 1A: The “SPR RBD Antibody Binding” graph of Part B of Figure 1 seems to present data about binding antibodies against BA.1, BA.2, and most probably BA.2.75. What are the variant selection criteria? What about data against the prototype WA1?

Response: SPR was used to determine the antibody binding against the receptor binding domain (RBD) of WA1, BA.1, BA.2.75 and BA.4/BA.5 spike proteins. This information is depicted in new Figure 1 as well as in the revised text.

Abstract: In addition, antibody binding to spike receptor binding domain (RBD) from WA1, BA.1, BA.2.75 and BA.4/BA.5 was determined using surface plasmon resonance (SPR).

Table 1: The upper age limit (241 months) must be 20 instead of 21 years.

Response: Upper age limit is now shown as 20 years as mentioned by the reviewer.

We included a statement: The study objective was to investigate the neutralizing activity of serum/plasma from children <20 years old hospitalized for COVID-19 or for MIS-C against circulating SARS-CoV-2 variants.

  1. 180: What are the selection criteria for the targeted variants?

Response: The criteria for strain selection was based on when specimens were obtained from children and young adults infected with SARS-CoV-2 between May 2020 and August 2022, a period during which the circulating SARS-CoV-2 strains were WA1, Alpha, Delta, and Omicron BA.1. We added newer Omicron strains that emerged following BA.1 and different circulating Omicron subvariants in mid-late 2023, when the samples were tested in neutralization assay. These included BA.2, BA.2.75, BA.4/BA.5, and highly transmissible BQ.1, BQ.1.1 and XBB.1 subvariants.

We have discussed it further in the discussion section.

  1. 342: Have pediatric specialists been involved in all the cases? To be clarified. (According to the CDC, in medical terms the age limit between children and adults is 17 years.)

Response: Yes. We have added the following information to the study design to clarify further in the methods section:

Study Design: Biospecimens were obtained from subjects enrolled at a pediatric hospital between May 2020, and August 2022.  Subjects were eligible for enrollment if they were <21 years of age and hospitalized with a diagnosis of either COVID-19 per SARS-CoV-2 antigen, PCR, or at-home test or MIS-C per the Center for Disease Control’s criteria. To ensure correct diagnostic group assignment, cases were reviewed by three pediatric specialists and agreement of the primary diagnosis reached.

Major revision

Overall: Each infecting variant is absolutely related to the produced specific antibodies and to the resulting cross reactions as well. What are the variants by which the research subjects were infected? In case there are no such details, it should be mentioned.

Response: The specimens were obtained from children and young adults infected with SARS-CoV-2 between May 2020 and August 2022, a period during which the circulating SARS-CoV-2 strains were WA1, Alpha, Delta, and Omicron BA.1.

We have added the following information in the results:

“Antibody neutralization activity of children and adolescents’ samples was evaluated against the infecting ancestral SARS-CoV-2 WA1, Alpha, Delta and the Omicron BA.1 strains, that circulated during May 2020 and August 2022.”

Reviewer 3 Report

Comments and Suggestions for Authors

1. This paper investigatesthe durability of neutralization capacity and level of binding antibody generated against the highly transmissible circulating Omicron subvariants following SARS-CoV-2 infection in children with acute COVID-19 and those diagnosed with multisystem inflammatory syndrome in children (MIS-C) in absence of vaccination.This study found that cross-neutralization capacity against recently emerged Omicron BQ.1, BQ.1.1 and XBB.1 variants was very low in children with either COVID-19 or MIS-C at all time-points,and suggest the need for vaccinations in children with prior COVID-19 or MIS-C to provide effective protection from emerging and circulating SARS-CoV-2 variants.This is a paper with great guiding value.

2. Suggest placing the "Methods" in the second part of the main text.

3. The sample size needs to be clearly defined in the methods section and abstract.

4. Please standardize the format of Table 1. The Age [months] media (IQR) is already shown in Table 1, so there is no need to list the Age [months] range. Delete media (IQR). Annotate the meanings of the numbers in the table and the numbers in () below the table.

5. Please standardize punctuation marks. For example, there should be no "." after the reference ().

There should be no "." after the title. For example: Line 110,214,362,382,411.

6.Is age, gender, and race balanced between the two groups (COVID-19 and MIS-C) in Table 1? Statistical analysis is required.

Comments on the Quality of English Language

Minor editing of English language required.

Author Response

We appreciate the reviewer’s comments to improve our manuscript. We have addressed all of them in the revised manuscript. We have provided point-by-point responses to the reviewer comments.

Reviewer 3:

  1. This paper investigates the durability of neutralization capacity and level of binding antibody generated against the highly transmissible circulating Omicron subvariants following SARS-CoV-2 infection in children with acute COVID-19 and those diagnosed with multisystem inflammatory syndrome in children (MIS-C) in absence of vaccination. This study found that cross-neutralization capacity against recently emerged Omicron BQ.1, BQ.1.1 and XBB.1 variants was very low in children with either COVID-19 or MIS-C at all time-points and suggest the need for vaccinations in children with prior COVID-19 or MIS-C to provide effective protection from emerging and circulating SARS-CoV-2 variants. This is a paper with great guiding value.

Response: We thank the reviewer for appreciating our study.

  1. Suggest placing the "Methods" in the second part of the main text.

Response: Methods have been moved to Section 2 before the Results section

  1. The sample size needs to be clearly defined in the methods section and abstract.

Response: Sample size (n=32 for COVID-19 and n=32 for MIS-C) has been added both to abstract and methods section.

  1. Please standardize the format of Table 1. The Age [months] media (IQR) is already shown in Table 1, so there is no need to list the Age [months] range. Delete media (IQR). Annotate the meanings of the numbers in the table and the numbers in () below the table.

Response: Table 1 has been revised as per reviewer’s suggestion.

  1. Please standardize punctuation marks. For example, there should be no "." after the reference ().

Response: Done where appropriate.

There should be no "." after the title. For example: Line 110,214,362,382,411.

Response: Done.

6.Is age, gender, and race balanced between the two groups (COVID-19 and MIS-C) in Table 1? Statistical analysis is required.

Response: Age, gender, and race were balanced between the two groups (COVID-19 and MIS-C). Statistical analysis did not identify any significant differences. Information has been added to revised Table 1.

Comments on the Quality of English Language

Minor editing of English language required.

Response: Manuscript has been edited and proofread for English language.

Round 2

Reviewer 1 Report

Comments and Suggestions for Authors

No further comments for the authors.

Comments on the Quality of English Language

None

Author Response

We thank the reviewer for appreciating our study.

The manuscript has been edited and proofread for English language.